Lung image segmentation with improved U-Net, V-Net and Seg-Net techniques

http://orcid.org/0000-0001-8159-360X Turk Fuat 1
http://orcid.org/0000-0003-1117-7736 Kılıçaslan Mahmut 2 m.kilicaslan@ankara.edu.tr
1 Department of Computer Engineering, Kirikkale University , Kırıkkale , Turkey
2 Department of Computer Technologies, Ankara University , Ankara , Turkey
Alatas Bilal
Electronic publication date: 2025 Feb 13
Publication date: 2025
Volume: 11
Electronic Location ID: e2700
Received 2024 Oct 24; Accepted 2025 Jan 22
Copyright: © 2025 Turk and Kılıçaslan
Copyright year: 2025
Copyright holder: Turk and Kılıçaslan
License: This is an open access article distributed under the terms of the Creative Commons Attribution License, which permits unrestricted use, distribution, reproduction and adaptation in any medium and for any purpose provided that it is properly attributed. For attribution, the original author(s), title, publication source (PeerJ Computer Science) and either DOI or URL of the article must be cited.
License URL: https://creativecommons.org/licenses/by/4.0/

Keywords: Lung segmentation, Improved U-Net, Improved V-Net, Seg-Net architecture, Medical image processing

Funding: The authors received no funding for this work.

==============================
Tuberculosis remains a significant health challenge worldwide, affecting a large population. Therefore, accurate diagnosis of this disease is a critical issue. With advancements in computer systems, imaging devices, and rapid progress in machine learning, tuberculosis diagnosis is being increasingly performed through image analysis. This study proposes three segmentation models based on U-Net, V-Net, and Seg-Net architectures to improve tuberculosis detection using the Shenzhen and Montgomery databases. These deep learning-based methods aim to enhance segmentation accuracy by employing advanced preprocessing techniques, attention mechanisms, and non-local blocks. Experimental results indicate that the proposed models outperform traditional approaches, particularly in terms of the Dice coefficient and accuracy values. The models have demonstrated robust performance on popular datasets. As a result, they contribute to more precise and reliable lung region segmentation, which is crucial for the accurate diagnosis of respiratory diseases like tuberculosis. In evaluations using various performance metrics, the proposed U-Net and V-Net models achieved Dice coefficient scores of 96.43% and 96.42%, respectively, proving their competitiveness and effectiveness in medical image analysis. These findings demonstrate that the Dice coefficient values of the proposed U-Net and V-Net models are more effective in tuberculosis segmentation than Seg-Net and other traditional methods.

Introduction

Medical image analysis has undergone a significant transformation with the emergence of deep learning techniques, which have demonstrated remarkable performance in various fields such as image classification, object detection, and semantic segmentation. In the medical domain, the segmentation and classification of lung images for the early detection of respiratory diseases, such as tuberculosis (TB), represent one of the most promising applications of deep learning. Tuberculosis remains a significant global health issue, and the number of people dying from the disease continues to increase each year (Lakhani & Sundaram, 2017; Chandra et al., 2020; Becker et al., 2018; Hwang et al., 2019). According to 2019 data, a total of 4.4 million people tested positive for the disease (Nafisah & Muhammad, 2024). Consequently, the detection of this disease has become a critical issue. Traditional lung image segmentation methods have faced challenges in accurately identifying lung regions, particularly in the presence of complex anatomical structures and pathological variations. In this context, the development of robust and effective deep learning-based approaches for lung image segmentation has become an urgent necessity, as they hold the potential to improve the accuracy and reliability of tuberculosis detection from chest X-ray images. Tuberculosis is a disease that can be detected through various methods (Gite, Mishra & Kotecha, 2023). X-ray images are frequently preferred for the diagnosis of this disease. Although X-ray imaging is cost-effective and easily accessible, manual segmentation requires significant human effort and is a challenging process (Dasanayaka & Dissanayake, 2021). Accurate diagnosis of TB is crucial, as misdiagnosis can lead to incorrect medications and treatments, which may have serious consequences (Degnan et al., 2019). Additionally, the early detection of tuberculosis is crucial both for treating the disease and for reducing the number of patients. For this reason, researchers are focusing on the autonomous segmentation of images obtained via X-ray. On the other hand, advancements in deep learning and image processing techniques enable the autonomous execution of such segmentation tasks. Machine learning and deep learning-based solutions have been proposed for various medical applications. A review of the literature reveals that deep learning-based approaches are frequently employed, particularly in the diagnosis of diseases such as brain tumours, lung diseases, and breast cancer. The successful outcomes of image classification and segmentation studies utilising deep learning continue to encourage researchers (Balamurugan & Balamurugan, 2024; Shome, Kashyap & Laskar, 2024; Sharma, Gupta & Shukla, 2024; Bansal, Gupta & Jindal, 2024).

In a recent study conducted in 2024, a complex segmentation network was employed to extract the region of interest from chest X-rays (CXR). In this study, experimental findings were obtained using various convolutional neural network (CNN) models. The classification performances of these CNN models were compared using three publicly available CXR datasets: Montgomery, Shenzhen, and Belarus (Nafisah & Muhammad, 2024). In another study, the authors proposed a deep learning (DL) architecture for multi-class classification of pneumonia, lung cancer, tuberculosis (TB), lung opacity, and more recently, COVID-19. Images from different categories were resized and processed with basic image preprocessing to make them suitable for classification. Experimental results were obtained using a three-block convolutional neural network (CNN) followed by a pre-trained model, VGG19. The study stands out with its performance findings, achieving approximately 96% accuracy (Alshmrani et al., 2023). On the other hand, the main disadvantages of the VGG model are its typically large network structure, the inclusion of a high number of parameters, and its high computational cost during training (Dahmane et al., 2021). One of the noteworthy approaches is the article conducted using the V-Net architecture. In their study, Wang, Men & Zhang (2023) used the V-Net architecture to test the performance of various enhanced models. The results indicated that the technique developed with V-Net was superior, achieving good segmentation performance and fast processing speed (Wang, Men & Zhang, 2023). In 2021, Zhao et al. (2021) proposed an automatic segmentation method. They developed an automatic approach to analyse tissue characteristics in chest CT images to assist radiologists in diagnosing COVID-19. The experimental results showed that the proposed segmentation method achieved a Dice similarity coefficient of 0.9796, a sensitivity of 0.9840, a specificity of 0.9954, and a mean surface distance error of 0.0318 mm (Zhao et al., 2021). Another convolutional neural network model used for effective segmentation is U-Net. There are numerous studies conducted using this technique (Ambesange, Annappa & Koolagudi, 2023; Bougourzi et al., 2023; Kumar et al., 2021; Agrawal & Choudhary, 2022; Zhang et al., 2024a; Hu et al., 2022). A study that proposed the 3D-UNet model for accurately segmenting air tubes from CT images produced successful results. A dataset of 300 CT images was used to evaluate its performance. Quantitative comparisons indicated that the proposed model achieved the best performance in terms of the Dice similarity coefficient (92.6%) and Intersection over Union (IoU; 86.3%), outperforming other state-of-the-art methods (Zhang et al., 2024b). A study investigating the impact of the U-Net architecture across different image groups was presented in 2023 by de la Sotta et al. (2024). The study aimed to assess the effect of attention-based mechanisms on segmenting different organs, such as the heart, lungs, clavicles, and ribs, in chest X-ray images, compared to the well-known U-Net architecture. Five variations of U-Net encoders were examined by replacing the U-Net encoder with simple residual structures consisting of ResNet 18, 34, and 50, Swin Transformer, and a single ResNet-50, before each U-Net encoder layer. The study achieved a high success rate, indicating that combining U-Net with ResNet produced superior performance (de la Sotta et al., 2024). A notable study is the approach proposed by Turk (2024), where the author analysed the performance of the U-Net model and its variations on the Shenzhen dataset. Experimental results indicated that the classic U-Net yielded lower accuracy for tuberculosis image segmentation than the RNGU-Net approach (Turk, 2024). Another model used for image segmentation is the Seg-Net deep learning approach (He et al., 2015; Badrinarayanan, Kendall & Cipolla, 2017). In 2017, the Seg-Net approach was introduced as an effective architecture for image segmentation. The proposed technique models a semantic pixel-based fully convolutional neural network. This structure consists of three layers: an encoder network, a corresponding decoder network, and a subsequent pixel-based classification layer. Its low computational cost provides a significant advantage (Badrinarayanan, Kendall & Cipolla, 2017).

The U-Net, V-Net, and Seg-Net segmentation models each offer significant advantages in specific areas. The U-Net segmentation model is particularly popular and effective in the fields of medical imaging and biomedical image segmentation. This method, specifically developed for medical and biomedical image segmentation, offers both high accuracy and efficient training time due to its symmetric architecture. Additionally, it can produce effective results even with small datasets and demonstrates strong performance on 2D images (Ronneberger, Fischer & Brox, 2015; Solórzano et al., 2021; Du et al., 2020). However, this model also has some disadvantages. Firstly, U-Net requires high computational power and memory, which becomes more pronounced when working with large and high-resolution images. V-Net, on the other hand, as a 3D version of U-Net, achieves remarkable results in 3D medical image segmentation. This model is optimised for working with 3D data, performs with high accuracy on large datasets, and provides a significant advantage, particularly in processing volumetric data (Al Husaini et al., 2020; Prathipati & Satpathy, 2024). However, when working with small datasets, V-Net also carries the risk of overfitting. Seg-Net, specifically designed for semantic segmentation, provides efficient and fast segmentation through its encoder-decoder structure. It stands out for its ability to perform segmentation without losing depth information, and it produces successful results by preserving details, particularly in high-resolution images. These three models offer powerful solutions for different segmentation needs and are widely used in the fields of image processing and computer vision.

Literature gap

Numerous studies have been conducted on lung segmentation, as outlined above. However, no study has been found that simultaneously compares the U-Net, V-Net, and Seg-Net models, nor one that introduces improved versions of the U-Net and V-Net models to the literature.

The primary contributions of this study to the scientific field include presenting the improved versions of the traditional U-Net and V-Net models for lung image segmentation. In this context, advanced preprocessing techniques have been employed on the images. The X-ray images in the dataset were reduced to a 512 × 512 resolution, and an adaptive enhancement architecture was introduced by using the wavelet transform coefficient obtained from each image in a diffusion filter. Finally, classic histogram equalisation was applied to the images to enhance their quality. Some attention mechanisms were added to the developed U-Net architecture, aiming to achieve more accurate and precise results. On the other hand, the proposed V-Net model incorporates non-local block structures, offering a different solution to the common bottleneck issue that results in a similarly high-performing model. Lastly, a comparative analysis was conducted between the U-Net and V-Net models, with Seg-Net also being compared as an alternative segmentation approach to the proposed models.

Materials and Methods

The U-Net, V-Net, and Seg-Net models, which are traditionally used and have shown success in medical image segmentation, have been enhanced. The three proposed methods were applied to the Montgomery and Shenzhen datasets, commonly utilised in lung segmentation research. Initially, the images from both datasets were combined. Subsequently, each image was resized to a 512 × 512 resolution, followed by image preprocessing steps. The improved images were then split, with 80% allocated for training and validation, while the remaining 20% were reserved for testing. After the segmentation processes were performed on the images, the results were evaluated using performance metrics. These steps are outlined in Fig. 1. The proposed model consists of four general steps: Data Loading, Training and Evaluation of Models, Model Results and Prediction and Visualization on Test Data. The pseudocode of the proposed algorithm is given in Table 1.

Figure 1 The architecture of the proposed model.

Table 1 Pseudocode for proposed algorithms.

pseudocode for proposed algorithms	
Data Loading	
      1:     shenzhen_data = load_images(shenzhen_path)	
      2:     montgomery_data = load_images(montgomery_path)	
      3:     combine_datasets(shenzhen_data, montgomery_data)	
      4:     train_data, val_data, test_data = load_data("combined_data ")	
      5:     return train_test_split(combined_data, test_size=0.2)	
Training and Evaluation of Models	
      6:     for model_type in ["U-Net","Prop.U-Net","V-Net","Prop. V-Net","Seg-Net"]:	
      7:            Training and evaluation: {model_type}	
      8:            model = create_model(model_type)	
      9:            models = {build_unet(input_shape),build_prop_unet(input_shape),build_vnet(input_shape), build_prop_vnet(input_shape),
                   "Seg-Net": build_segnet(input_shape)}	
Model Results	
      10:    history, evaluation = train_and_evaluate_model(model, train_data, val_data)	
      11:    ResultofModel: {evaluation}") metrics=["IOU", "dice_coefficient"])	
Prediction and Visualization on Test Data	
      12:    predict_and_visualize(model, test_data)	

Dataset

The Montgomery (https://www.kaggle.com/datasets/raddar/tuberculosis-chest-xrays-montgomery) and Shenzhen (https://www.kaggle.com/datasets/raddar/tuberculosis-chest-xrays-shenzhen) image public datasets are two important datasets frequently used in segmentation studies. Numerous studies have been conducted using these datasets (Sharma, Gupta & Shukla, 2024; Kotei & Thirunavukarasu, 2024; Ammar, Gasmi & Ltaifa, 2024; Balamurugan & Balamurugan, 2024; Morís et al., 2022). The Montgomery dataset is a labelled collection of frontal chest X-ray images. It contains 138 X-ray images, of which 80 are free from any disease, while 58 show tuberculosis infection. This dataset, which includes lung segmentation masks in DICOM format, was created by the Department of Health and Human Services of Montgomery. Similarly, the Shenzhen dataset is another labelled collection of frontal chest X-ray images. This dataset contains 662 X-ray images, of which 326 represent inactive cases and 336 show active tuberculosis infections. The dataset is provided in JPEG format (Chandra et al., 2020; Stirenko et al., 2018; Nafisah & Muhammad, 2024; Jaeger et al., 2014). Both databases can be downloaded from https://openi.nlm.nih.gov/. The data are randomly distributed into two groups, 80% allocated for training and 20% allocated for testing. The “train_test_split” command in the Python sklearn library was imported for the distribution process. The results in this section are the average of the five-fold cross-validation results obtained from the training dataset. The validation coefficient was taken as 5 and it was aimed to increase the reliability of the study by using different data groups for each test. The numbers related to the distribution rates of the data are shown in the Table 2.

Table 2 Shenzhen and Montgomery dataset distribution (train/val-test).

	Tuberculosis	Normal	Total image	
Montgomery dataset	58	80	138	
Shenzen dataset	336	326	662	
Merging dataset	394	406	800	
Train-Val/Test
(Total)	800	Val-group	Test group	
Train/Val-Test
(k = 1)	640-160	1,2,3,4	5	
Train/Val-Test
(k = 2)	640-160	1,2,3,5	4	
Train/Val-Test
(k = 3)	640-160	1,2,4,5	3	
Train/Val-Test (k = 4)	640-160	1,3,4,5	2	
Train/Val-Test (k = 5)	640-160	2,3,4,5	1	

Image preprocessing

The image preprocessing stage is critical for segmentation tasks. In segmentation processes, ensuring that images are noise-free and regions are clearly defined leads to more successful outcomes. In this study, an effective image processing method was applied to the images in both datasets. First, each image was resized to a 512 × 512 resolution. Then, the wavelet transform coefficients were calculated for each image in horizontal, vertical, and diagonal directions, and their averages were taken. Subsequently, the obtained coefficients were used as conductivity parameters in the anisotropic diffusion filter. This helped to clean unwanted pixels that may be present in the images. In other words, an adaptive filter was applied using the coefficients specific to each image. Finally, histogram equalisation was performed to make the regions more distinct. The 512 × 512 resolution of the raw images was chosen to reduce the computational cost of the model while ensuring that visual details are adequately preserved. Adaptive image filtering was applied to prevent the loss of important details, especially in noisy data, and to reduce noise so that the model learns more stably. Histogram equalization aimed to facilitate the classification process by increasing the contrast of the images and to enable the model to perform better on images with different brightness levels. Each of these steps was evaluated experimentally and the effects of the results on accuracy and generalization performance were analyzed in detail in the study.

Segmentation models

There are numerous deep learning algorithms used for medical image segmentation. However, most algorithms are generally structured in the form of U-Net and its derivatives. In addition to these methods, the V-Net model has been introduced as an alternative to overcome bottleneck issues in segmentation processes. On the other hand, the Seg-Net model has recently gained attention as a popular approach. A key advantage of the proposed methods is their adaptability in terms of resource utilisation. Furthermore, they are designed to address fundamental segmentation challenges effectively.

Proposed U-Net

Figure 2 illustrates the architecture of the proposed U-Net model. The classic U-Net architecture consists of three sections: encoder, bottleneck, and decoder. In the encoder part, convolutional layers are applied sequentially to the input image. Max-pooling layers are used to reduce the size of the feature maps. The bottleneck phase is the intermediate layer that connects the encoder and decoder, where the smallest but deepest feature maps are obtained. In the decoder part, upsampling operations are applied to increase the size of the feature maps. After upsampling, convolutional layers are used again to capture more details. Finally, skip connections are added to transfer the feature maps at the same level from the encoder to the corresponding layers in the decoder. This minimises detail loss and improves segmentation results.

Figure 2 Proposed U-Net architecture.

In the proposed architecture, two additional blocks are introduced, differing from the classic U-Net architecture. The first is the Attention U-Net block, where attention mechanisms are added to the classic U-Net to achieve more precise and accurate segmentation results. This block helps better capture the region of interest and improves segmentation accuracy. Attention layers compute the importance of each pixel, assigning more weight to significant pixels, thus creating a weighted structure. The filtered data is then passed to the decoder after filtering the feature maps from the encoder.

Secondly, residual block structures have been used. The main objective here is to enhance the U-Net architecture by adding residual connections to each convolution block, enabling the transfer of feature maps to deeper layers and reducing gradient loss. A residual block consists of two or more convolutional layers with a skip connection added between the input and output. This skip connection allows the direct addition of the input to the output. Residual blocks are employed only in the encoder phase to extract more features, but they are not used in the decoder phase, allowing the model to operate faster.

Proposed V-Net

Figure 3 shows the architectural structure of the proposed V-Net model. V-Net is a deep learning model specifically designed for medical image segmentation. It is primarily used for the segmentation of three-dimensional medical images, such as magnetic resonance imaging (MRI) and computed tomography (CT). The V-Net architecture is fundamentally similar to U-Net but differs in the structure of convolutional layers and the preference for downsampling steps instead of pooling layers. The reduction in the number of volumetric features occurs due to these differences, which aims to overcome bottleneck issues.

Figure 3 Proposed V-Net architecture.

In addition to these parameters, the proposed V-Net model includes non-local block structures. During the bottleneck phase, the non-local block architecture is used to model the relationship between each position in the input feature map and all other positions, enabling the learning of long-range dependencies. In the fundamental steps of these blocks, the input feature map is embedded into a lower-dimensional space using a specific function. A proximity matrix is then calculated using these embedded features, which indicates the relationship between each position and others. In the Weighted Sum step, the proximity matrix is used to compute the weighted sum of all other positions, which is then combined with the input feature map. By leveraging non-local blocks, relationships between features located in distant regions are modelled, resulting in more accurate and precise segmentation. This approach enhances performance in the segmentation of complex structures and fine details.

Proposed Seg-Net

Figure 4 shows the architectural structure of the proposed Seg-Net model. Seg-Net is a deep learning architecture used for image segmentation, particularly in semantic segmentation tasks. Seg-Net is primarily based on an encoder-decoder structure and is used to transform an input image into an output image that matches pixel by pixel. Unlike the U-Net architecture, Seg-Net applies batch normalisation after each convolutional layer in both the encoder and decoder stages. Seg-Net is typically configured as a modified version of the VGG16 architecture.

Figure 4 Proposed Seg-Net architecture.

Experimental results and discussions

In this study, segmentation methods developed using the proposed U-Net, V-Net and Seg-Net models were tested on Schenzen and Montgomery databases. The aim is to evaluate the performance of these three different models and compare their effectiveness in medical image segmentation. During the training stages of the models, Adam optimizer, ReLU activation function was used, learning rate was set to 1e−3, step sizes in the convolutional block were set to 2 * 2 and filter sizes were set to 3 * 3. In addition, the hyperparameters used are shown in detail in the Table 3.

Table 3 Hyperparameters of conventional and proposed segmentation models.

Model	Filter size	Stride	Max pool	Learning rate	Batch size	Optimizer	Activation func.	Epoch	
U-Net	3 * 3	2 * 2	2 * 2	1e−3	32	Adam	ReLU	50	
V-Net	3 * 3	2 * 2	–	1e−3	32	Adam	ReLU	50	
Seg-Net	3 * 3	2 * 2	2 * 2	1e−3	32	Adam	ReLU	50	
Proposed U-Net	3 * 3	2 * 2	2 * 2	1e−3	32	Adam	ReLU	50	
Proposed V-Net	3 * 3	2 * 2	–	1e−3	32	Adam	ReLU	50	

Performance metrics

The segmentation of medical images is a process that plays a critical role in medical diagnosis and treatment planning. In this process, various performance metrics are used to evaluate the accuracy and efficiency of the segmentation. These metrics allow for an objective assessment of the models’ success by measuring different aspects of segmentation performance. Typically, these metrics are calculated by comparing the segmentation results with ground truth masks that are considered as reference standards (Ronneberger, Fischer & Brox, 2015; Badrinarayanan, Kendall & Cipolla, 2017; Milletari, Navab & Ahmadi, 2019). In this study, commonly used performance metrics, such as accuracy, IoU, precision, specificity, and the Dice coefficient, were employed to evaluate the effectiveness of the segmentation models. These metrics not only indicate how successfully the segmentation was performed but also allow for the comparison of different models. The formulations of these metrics are as follows:

(1) Dicecoefficient=2×TP/(2×TP+FN+FP)

(2) IoU(JaccardIndex)=TP/(TP+FN+FP)

(3) Accuracy=(TP+TN)/(TP+FN)+(FP+TN)

(4) Precision=TP/(TP+FP)

(5) Recall=TP/(TP+FN)

(6) Loss=−(Dicecoefficient).

Performance evaluation for proposed U-Net

The training and validation results of the proposed U-Net model are presented with various metrics in Table 4. Additionally, Figs. S1 and S2 illustrate the curves of loss, accuracy, IoU, and Dice coefficient parameters of the proposed method on the datasets. When the performance of the proposed model is evaluated based on various metrics, rapid improvement is observed in both the training and validation sets. The loss function decreases rapidly at first, and then the training and validation losses stabilise at similar levels, indicating that the model has acquired generalisation ability without overfitting. The accuracy metric reveals that both curves approach nearly 100%, demonstrating that the model performs almost perfectly on both the training and validation sets. Similarly, high values are also observed in metrics such as the Jaccard Index (IoU) and Dice coefficient, which confirms that the model successfully predicts the overlap between the predicted and actual segmentation masks and exhibits strong performance in terms of segmentation accuracy. However, the traditional U-Net model achieved remarkable results in the training phase, with IoU and Dice coefficient metrics reaching approximately 0.97 and 0.98, respectively. On the other hand, in the proposed U-Net model, these results are around 0.94 and 0.97. Therefore, it can be understood that the traditional U-Net slightly outperforms the proposed model. Nevertheless, the proposed model has produced better results in terms of sensitivity and precision compared to the U-Net model. This clearly demonstrates that the model is more successful in identifying true positives and exhibits lower rates of false positives.

Table 4 Conventional and proposed U-Net best performing training and validation results with various evaluation metrics.

Model	Loss	Dice	IoU	Recall	Precision	Accuracy	
U-Net training	−0.9842	0.9842	0.9699	0.9808	0.9869	0.9918	
U-Net validation	−0.9398	0.9398	0.8977	0.9422	0.9576	0.9749	
Proposed
U-Net training	−0.9708	0.9708	0.9449	0.9658	0.9760	0.9853	
Proposed
U-Net validation	−0.9605	0.9607	0.9265	0.9525	0.9694	0.9800	

In general, while the U-Net model provides better results in specific metrics, the proposed model shows superiority in metrics such as sensitivity and precision. This suggests that the proposed model may be more robust in some cases and capable of making more accurate predictions in specific scenarios. Both models exhibit high accuracy and overall performance, yet the proposed technique offers comparable performance to the U-Net model and stands out with superior performance in certain metrics.

Figure S3 presents the original images, the corresponding truth masks, and the predicted masks resulting from the model’s training and validation processes. It has been observed that the predicted masks perform well, even in fine details; however, in some challenging cases, erroneous predictions were made.

Performance evaluation for proposed V-Net

The training and validation results of the proposed model are highlighted in Figs. S4 and S5. Similarly, Table 5 provides a performance evaluation based on different metrics. Upon reviewing the aforementioned figures, it can be observed that the loss and accuracy values for both training and validation processes are quite close to each other. Figure S6 presents the original images, the corresponding ground truth masks, and the predicted masks resulting from the model’s training and validation processes. When comparing the performance of the proposed V-Net architecture with that of the classical V-Net across different metrics, distinct strengths and limitations of both models during the training and validation processes become evident. The differences between the metrics of the classical V-Net and the proposed model are noteworthy when examining the training and validation curves. Evaluating the loss and accuracy curves shows that both models quickly reach low loss values, while the validation loss closely follows the training loss. However, the validation accuracy of the classical V-Net stabilises at a slightly lower value, indicating that the proposed V-Net model performs slightly better on the validation set. A similar trend is observed in the IoU and Dice coefficient metrics. While the classical V-Net achieves high values during training, its performance decreases somewhat on the validation set. In contrast, the validation performance of the proposed V-Net model remains closer to its training performance, clearly demonstrating that the model possesses stronger generalisation capabilities. When examining the loss values in Table 5, it can be observed that the classical V-Net achieves lower values during training but loses this advantage in the validation phase. The proposed model, on the other hand, provides more consistent loss values across both training and validation processes, indicating that the model undergoes a more balanced learning process. Although the classical V-Net demonstrates a higher Dice coefficient during training, its performance decreases on the validation set. In the proposed model, this decline is less pronounced, suggesting that the proposed model offers a more reliable performance on the validation set. The recall and precision metrics indicate that the proposed model performs better. This shows that the model is not only superior in detecting true positives but also in minimising false positives. Overall, the proposed V-Net architecture exhibits a more balanced performance compared to the classical V-Net. While the classical V-Net shows higher performance during training, its performance drops in the validation set, which suggests that the model may be prone to overfitting. In contrast, the proposed V-Net model presents more consistent and reliable results across both training and validation sets. This indicates that the model’s generalisation ability and practical applicability are higher. These findings suggest that the proposed model could be more effective in clinical or industrial applications.

Table 5 Conventional and proposed V-Net best performing training and validation results with various evaluation metrics.

Model	Loss	Dice	IoU	Recall	Precision	Accuracy	
V-Net training	−0.9848	0.9848	0.9717	0.9823	0.9873	0.9922	
V-Net validation	−0.9389	0.9389	0.8909	0.9340	0.9543	0.9730	
Proposed
V-Net training	−0.9731	0.9708	0.9449	0.9658	0.9760	0.9853	
Proposed
V-Net validation	−0.9605	0.9607	0.9265	0.9525	0.9694	0.9800	

Performance evaluation for proposed Seg-Net

In this section, no new blocks have been added to Seg-Net. Only the model’s hyperparameters were adjusted as specified in “Experimental Results and Discussions”. The aim was to design the model at a basic level and compare it with the U-Net and V-Net models. When examining the training and validation curves in Figs. S7 and S8, significant fluctuations in loss and accuracy values can be observed. These fluctuations indicate that the model experiences instability during the learning process and encounters various challenges. The sudden drops, particularly in the loss function and validation, may suggest that the model faces issues such as overfitting or underfitting at certain stages. Such fluctuations negatively affect the model’s generalisation ability in segmentation problems. However, the reduction of these fluctuations in the later stages of the training process indicates that the model enters a more stable learning phase and begins to deliver more consistent results on the validation set.

Table 6 presents the performance of the proposed Seg-Net model. When compared to the training and validation results of the U-Net and V-Net models, the results are somewhat lower. However, the model still proves to be successful, particularly considering its simple structure and segmentation outcomes. Additionally, the performance metrics calculated for classification are also noteworthy. These results are remarkable for a model with simple implementation and basic hyperparameters, demonstrating that the Seg-Net model remains competitive despite its simplicity.

Table 6 Conventional and proposed Seg-Net best performing training and validation results with various evaluation metrics.

Model	Loss	Dice	IoU	Recall	Precision	Accuracy	
Proposed
Seg-Net training	−0.9612	0.9612	0.9529	0.9701	0.9776	0.9868	
Proposed
Seg-Net validation	−0.9494	0.9494	0.9293	0.9550	0.9662	0.9803	

In Fig. S9, the original images, their corresponding ground truth masks, and the predicted masks generated by the model after training and validation are displayed. The predicted masks demonstrate a performance that is quite respectable when compared to the U-Net and V-Net models. However, given that U-Net and V-Net are specifically optimised for segmentation tasks, it is understandable that they produce superior results.

Segmentation techniques test results on all models

In this section, the test results of the proposed models for the study are compared. When reviewing Table 7, it can be observed that the Dice coefficient and IoU values of the proposed U-Net and V-Net models are nearly identical. The recall, precision, and accuracy values are also quite close across all models, with the Seg-Net model showing slightly better performance. As these results suggest, the proposed models outperform the baseline models in terms of segmentation metrics. Additionally, it indicates that the Seg-Net model may be more successful in classification tasks.

Table 7 Comparison of test performances of the proposed models.

Model	Loss	Dice	IoU	Recall	Precision	Accuracy	
U-Net testing	−0.9560	0.9560	0.9166	0.9513	0.9610	0.9787	
Proposed
U-Net testing	−0.9643	0.9643	0.9326	0.9618	0.9669	0.9824	
V-Net testing	−0.9531	0.9531	0.9118	0.9467	0.9599	0.9769	
Proposed
V-Net testing	−0.9642	0.9642	0.9327	0.9592	0.9692	0.9824	
Seg-Net testing	−0.9551	0.9551	0.9395	0.9670	0.9661	0.9834	

In Table 8, the performance of various models is represented by the Dice coefficient and accuracy metrics. The Dice coefficient and accuracy metric indicate how successfully a model performs classification or segmentation tasks. When examining the table, the highest Dice value of 0.977 (Liu et al., 2022) which also achieves the highest accuracy value of 0.9890. Looking at the proposed models, the U-Net model shows strong performance with a Dice value of 0.9643 and a high accuracy of 0.9824. This demonstrates that U-Net is successful in segmentation tasks and can compete with models in the literature. Similarly, the proposed V-Net model shows nearly identical performance to U-Net in terms of both Dice and accuracy values (Dice: 0.9642, Acc: 0.9824), proving that V-Net also offers strong performance and can be used effectively in segmentation problems. Although the proposed Seg-Net model shows good performance with a Dice value of 0.9551 and an accuracy of 0.9834, its performance is slightly lower compared to U-Net and V-Net. When compared to models in the literature, the proposed models are found to be highly competitive, particularly in terms of accuracy. There are recent approaches and are remarkable achievements researches in Table 8 (Alam et al., 2024; Boudoukhani et al., 2024; Slimani & Bentourkia, 2024; Ammar, Gasmi & Ltaifa, 2024). The results of these references compare the performance of several methods on the Shenzhen and Montgomery datasets. Among the methods in the literature, the article (Boudoukhani et al., 2024) stands out with a Dice value of 0.9742 on the Montgomery dataset, while other method (Slimani & Bentourkia, 2024) shows the adequate performance with an Dice of 0.9570. On the other hand, one of the current approaches provides accuracy values for Montgomery and Shenzhen datasets, but the accuracy value in Shenzhen (0.96) is higher than Montgomery (0.94) (Ammar, Gasmi & Ltaifa, 2024). The proposed methods (U-Net, V-Net and Seg-Net) are generally quite satisfactory. The proposed U-Net has an accuracy of 0.9824 and Dice value of 0.9643, which is close to the approach with an accuracy of 0.9868 (Boudoukhani et al., 2024). The V-Net approach has a performance very close to U-Net. On the other hand, Seg-Net has a slightly lower performance than the other two proposed methods with an accuracy value of 0.9551, but it has achieved a high accuracy with an accuracy value of 0.9834. Overall, the proposed U-Net and V-Net models perform competitively with many robust models in the literature and provide viable solutions for segmentation problems. In conclusion, the proposed models have achieved strong results, especially in terms of accuracy. Therefore, they are promising for future studies.

Table 8 Performance evaluations of various approaches with proposed models.

Study	Dice	Accuracy	Dataset	
Ronneberger, Fischer & Brox (2015)	0.9029	0.9036	JSRT	
Montgomery	
Milletari, Navab & Ahmadi (2019)	0.9667	0.9804	Montgomery	
Shenzhen	
Rajaraman et al. (2018)	–	0.9170	Own	
Arvind et al. (2023)	0.9187	0.9387	JSRT	
Montgomery	
Shenzhen	
Liu et al. (2022)	0.977	0.9890	JSRT	
Montgomery	
Showkatian et al. (2022)	–	0.8725	Maryland	
Montgomery	
Shenzhen	
Oktay et al. (2018)	0.9723	0.9816	Own	
Huang et al. (2020)	0.9290	0.9187	ISBI LiTS 2017	
Alam et al. (2024)	0.9413	0.8916	Shenzhen	
Boudoukhani et al. (2024)	0.9742	0.9868	Montgomery	
0.9607	0.9798	Shenzhen	
Slimani & Bentourkia (2024)	0.9570	0.9790	Montgomery	
Ammar, Gasmi & Ltaifa (2024)	–	0.94	Montgomery	
–	0.96	Shenzhen	
Proposed U-Net	0.9643	0.9824	Montgomery
Shenzhen	
Proposed V-Net	0.9642	0.9824	Montgomery
Shenzhen	
Proposed Seg-Net	0.9551	0.9834	Montgomery
Shenzhen	

Table 9 presents statistical information about the training time, resource utilization and epoch/batch size parameters for the proposed U-Net, V-Net and Seg-Net models. In many segmentation problems, resource utilization and training time increase in parallel with the parameters used for a better segmentation process. Therefore, Table 8 shows that there is an increase in the processing time of the proposed models. However, considering the effectiveness of the related models, it can be said that the time increase is negligible. Therefore, the applicability and success of the proposed architectures are remarkable.

Table 9 Statistical information about the proposed models.

Model	Training time (min)	Resource utilization	Epoch/batch size	
U-Net	26	RTX4060	50/32	
Prop.U-Net	32	RTX4060	50/32	
V-Net	28	RTX4060	5032	
Prop.V-Net	33	RTX4060	50/32	
Seg-Net	35	RTX4060	50/32	

Conclusions

In this article, lung segmentation models developed based on U-Net, V-Net, and Seg-Net architectures were presented, with a focus on detecting tuberculosis in chest X-ray images. The proposed models were applied to the Shenzhen and Montgomery datasets, and the results confirmed that our models outperform existing methods. Additionally, preprocessing steps such as adaptive filtering and histogram equalisation improved image quality, further enhancing model performance. The proposed U-Net and V-Net models are more successful than the Seg-Net model (0.9551) with Dice coefficient values of 0.9643 and 0.9642 respectively. Although the Seg-Net model showed slightly lower performance compared to U-Net and V-Net, it still produced competitive results and demonstrated high accuracy, particularly in classification tasks. Even this small difference is very valuable for segmentation problems. These findings suggest that the proposed models, especially U-Net and V-Net, are promising tools for automatic TB detection and could support healthcare professionals in clinical applications. The primary drawbacks of this and similar studies include low performance values in tuberculosis segmentation, insufficient utilization of system resources (such as GPU and CPU), and implementation challenges due to complex architectures. An important limitation of this study is that, although the proposed models yield successful results, they must be tested on different datasets. Within the scope of the proposed methodology, it has been observed that performance values improved to some extent, while resource usage did not impose a significant burden. Future studies are expected to comprehensively evaluate these models using datasets collected from various regions. We remain optimistic that the proposed methodology can deliver effective results with reduced data and resource usage.

Supplemental Information

Supplemental Information 1 Source code.

Supplemental Information 2 Computing infrastructure.

Supplemental Information 3 Justification for model type used.

Supplemental Information 4 Assessment justification.

Supplemental Information 5 Readme.

Supplemental Information 6 Performance of proposed U-Net on training and validation dataset: left:loss right: Accuracy.

Supplemental Information 7 Performance of proposed U-Net on training and validation dataset: left:IoU right: Dice Coefficient.

Supplemental Information 8 Visual evaluations: left: original images, center: true masks; right: predicted masks.

Supplemental Information 9 Performance of proposed V-Net on training and validation dataset: left:loss right:accuracy.

Supplemental Information 10 Performance of proposed V-Net on training and validation dataset: left:IoU right:Dice Coefficient.

Supplemental Information 11 Visual evaluations: left: originals images, center: true masks, right: predicted masks.

Supplemental Information 12 Performance of proposed Seg-Net on training and validation dataset: left:loss right: accuracy.

Supplemental Information 13 Performance of proposed Seg-Net on training and validation dataset: left:IoU right:Dice Coefficient.

Supplemental Information 14 Visual evaluations: left: originals images, center: true masks, right: predicted masks.

Additional Information and Declarations

Competing Interests

The authors declare that they have no competing interests.

Author Contributions

Fuat Turk conceived and designed the experiments, performed the experiments, analyzed the data, performed the computation work, authored or reviewed drafts of the article, and approved the final draft.

Mahmut Kılıçaslan conceived and designed the experiments, performed the experiments, prepared figures and/or tables, authored or reviewed drafts of the article, and approved the final draft.

Data Availability

The following information was supplied regarding data availability:

The raw measurements are available in the Supplemental Files.

Access to the Shenzen and Montgomery datasets from the National Library of Medicine can be requested at: https://openi.nlm.nih.gov/faq#faq-tb-coll.

Montgomery Dataset: https://www.kaggle.com/datasets/raddar/tuberculosis-chest-xrays-montgomery

Shenzen Dataset: https://www.kaggle.com/datasets/raddar/tuberculosis-chest-xrays-shenzhen.

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
