# Peer review of "Lung image segmentation with improved U-Net, V-Net and Seg-Net techniques"

_PeerJ Computer Science, doi:10.7717/peerj-cs.2700_

## Round 0.1 · original submission · Major Revisions

Hi,
You need to address all the concerns of the expert reviewers.
/Khursheed

Reviewer 1 ·

Basic reporting

No comment

Experimental design

No comment

Validity of the findings

No comment

Additional comments

Very well written.

Cite this review as

·

Basic reporting

1. Describe dataset features in more details and its total size and size of (train/test) as a table.
2. Pseudocode / Flowchart and algorithm steps need to be inserted.
3. Time spent need to be measured in the experimental results.
4. Limitation and Discussion Sections need to be inserted.
5. Comparison with similar studies on a similar dataset need to be inserted (with references).
6. All metrics need to be calculated in the experimental results.
7. The parameters used for the analysis must be provided in table
8. The architecture of the proposed model must be provided
9. Address the accuracy/improvement percentages in the abstract and in the conclusion sections, as well as the significance of these results.
10. The authors need to make a clear proofread to avoid grammatical mistakes and typo errors.
11. Add future work in last section (conclusion) (if any)
12. Enhance the clarity of the Figures by improving their resolution.

Experimental design

Comparison with similar studies on a similar dataset need to be inserted (with references).

Validity of the findings

All metrics need to be calculated in the experimental results.

Reviewer 3 ·

Basic reporting

All comments are included in detail in the fourth section.

Experimental design

All comments are included in detail in the fourth section.

Validity of the findings

All comments are included in detail in the fourth section.

Additional comments

Review Report for PeerJ Computer Science
(Lung image segmentation with improved U-Net, V-Net and SegNet techniques)

1. Within the scope of the study, various lung image segmentation operations were performed based on three different deep learning based models using two different open source datasets.

2. In the introduction, the importance of the subject, medical image analysis, studies in the literature on the subject and the parts that this study has the potential to present to the literature are mentioned in detail and sufficiently.

3. Shenzhen and Montgomery datasets shared as open source from the Kaggle platform were preferred for the study. Although there are many different datasets in the literature that can be used to solve this problem, it is positive to use both these datasets and to use the proposed models in two different datasets instead of sticking to a single dataset.

4. Passing the dataset through various preprocessing steps instead of using it raw increased the quality of the study. However, when both the preferred image resolution and the preprocessing steps used are examined in detail, it should be detailed exactly how it was determined and whether different experiments were made. It is also recommended to observe what kind of positive/negative effect the proprocessing steps have on the result.

5. Since the segmentation results are very dependent on the dataset distribution and the test dataset, how the distribution is determined should be detailed.

6. The proposed models have a certain level of originality. In addition, the evaluation metric types and the obtained results reveal the impact of the study. However, for a more detailed analysis of the results and for the proposed models to come to the fore, it is recommended to compare the superiority of this study by performing segmentation with some of the state-of-the-art deep learning based models in the literature.

As a result, although the quality of the study is high, the sections listed above should be addressed completely.

Cite this review as

---

## Round 0.2 · accepted · Accept

Dear Authors,

One of the preceding reviewers did not respond to the invitation to review the manuscript. One of them has already accepted the first version, and this latest revision has been accepted by another reviewer. I have already assessed the revision myself, and I can confirm that your manuscript seems sufficiently improved. The paper now seems ready for publication.

Best wishes,

Reviewer 3 ·

Basic reporting

All comments are included in detail in the fourth section.

Experimental design

All comments are included in detail in the fourth section.

Validity of the findings

All comments are included in detail in the fourth section.

Additional comments

Thank you for the revision. The responses to the my reviewer comments and the changes made to the paper are suitable. For this reason, I recommend that the paper be accepted. I wish the authors success in their future papers. Best regards.

Cite this review as